# The Molecular Epidemiology of HIV-1 in Russia, 1987–2023: Subtypes, Transmission Networks and Phylogenetic Story

**DOI:** 10.3390/pathogens14080738

**Published:** 2025-07-26

**Authors:** Aleksey Lebedev, Dmitry Kireev, Alina Kirichenko, Ekaterina Mezhenskaya, Anastasiia Antonova, Vyacheslav Bobkov, Ilya Lapovok, Anastasia Shlykova, Alexey Lopatukhin, Andrey Shemshura, Valery Kulagin, Aleksei Kovelenov, Alexandra Cherdantseva, Natalia Filoniuk, Galina Turbina, Alexei Ermakov, Nikita Monakhov, Michael Piterskiy, Aleksandr Semenov, Sergej Shtrek, Aleksej Sannikov, Natalia Zaytseva, Olga Peksheva, Aleksandr Suladze, Dmitry Kolpakov, Valeriia Kotova, Elena Bazykina, Vasiliy Akimkin, Marina Bobkova

**Affiliations:** 1I. Mechnikov Research Institute for Vaccines and Sera, 105064 Moscow, Russia; vbobkov@gmail.com (V.B.); mrbobkova@mail.ru (M.B.); 2Central Research Institute of Epidemiology, 111123 Moscow, Russia; kotova-kirichenko@mail.ru (A.K.); i_lapovok@mail.ru (I.L.); murzakova_a.v@mail.ru (A.S.); a.lopatukhin@gmail.com (A.L.); vgakimkin@yandex.ru (V.A.); 3Gamaleya National Research Center for Epidemiology and Microbiology, 123098 Moscow, Russia; belokopytova.01@mail.ru (E.M.); anastaseika95@mail.ru (A.A.); 4Federal Budget Public Health Institution “Clinical Center of HIV/AIDS Treatment and Prevention” of the Ministry of Health of Krasnodar Region, 350000 Krasnodar, Russia; shemsh@mail.ru (A.S.); aidskuban@mail.ru (V.K.); 5Federal State Budgetary Educational Institution of Higher Education “Kuban State Medical University” of the Ministry of Health of the Russian Federation, 350063 Krasnodar, Russia; 6AIDS Centre of Leningrad Region, 191124 St. Petersburg, Russia; akovelenov@mail.ru (A.K.); alexcerdinfdoctor@gmail.com (A.C.); 7Lipetsk Regional Center of Infectious Diseases, 398043 Lipetsk, Russia; natafilonyuk@mail.ru (N.F.); galinaturbina@yandex.ru (G.T.); 8St. Petersburg City AIDS Center, 190020 St. Petersburg, Russia; ermakovspb@mail.ru (A.E.); kitt_898989@mail.ru (N.M.); 9Federal Budgetary Institution of Science “Federal Scientific Research Institute of Viral Infections “Virome” Federal Service for Surveillance on Consumer Rights Protection and Human Wellbeing”, 620030 Ekaterinburg, Russia; piterskiy_mv@niivirom.ru (M.P.); alexvsemenov@gmail.com (A.S.); 10Omsk Research Institute of Natural-Focal Infections, 644080 Omsk, Russia; studi1990@mail.ru (S.S.); sannikov.1.96@mail.ru (A.S.); 11Omsk State Medical University, 644099 Omsk, Russia; 12Academician I.N. Blokhina Nizhny Novgorod Scientific Research Institute of Epidemiology and Microbiology, 603950 Nizhny Novgorod, Russia; vtashca@mail.ru (N.Z.); peolinn@mail.ru (O.P.); 13Rostov Research Institute of Microbiology and Parasitology, 344000 Rostov-on-Don, Russia; sualrostov@mail.ru (A.S.); dimakolpakov@mail.ru (D.K.); 14Khabarovsk Research Institute of Epidemiology and Microbiology of the Rospotrebnadzor, 680610 Khabarovsk, Russia; kotova.valeriya@mail.ru (V.K.); dvaids@mail.ru (E.B.)

**Keywords:** HIV-1, subtypes, phylodynamic analyses, transmission clusters, Russia

## Abstract

Regional HIV-1 epidemics are evolving with distinct patterns in transmission routes, subtype distribution, and molecular transmission cluster (MTCs) characteristics. We analyzed 9500 HIV-1 cases diagnosed over 30 years using phylogenetic and network methods, integrating molecular, epidemiological, demographic, and behavioral data. Subtype A6 remains dominant nationally (80.6%), followed by 63_02A6 (7.9%), subtype B (5.6%), 02_AG_FSU_ (1.2%), 03_A6B (0.7%), and 14/73_BG (0.6%). Non-A6 infections were more common among males (OR 1.51) and men who have sex with men (OR 7.33). Network analysis identified 421 MTCs, with 256 active clusters. Clustering was more likely among young individuals (OR: 1.31), those not receiving antiretroviral therapy (OR: 2.70), and injecting drug users (OR: 1.28). Non-A6 subtypes showed a higher likelihood of clustering. Phylogenetic analysis revealed that local clusters of the major subtypes originated between the late 1970s (subtype B) and the mid-2000s (63_02A6) with links to populations in Eastern Europe, Central Asia (subtypes A6, 63_02A6, 02_AG_FSU_, 03_A6B), and Western Europe and the Americas (subtype B, 14/73_BG). These findings indicate a complex, evolving regional epidemic transitioning from subtype A6 dominance to a more diverse mix of subtypes. The ability of non-A6 subtypes to form active MTCs suggests their establishment in the local population.

## 1. Introduction

With the growing number of available HIV-1 nucleotide sequences, studies analyzing viral genomic data are becoming increasingly informative and impactful. While viral sequencing was initially performed primarily to assess drug resistance and for basic research, its applications have now expanded significantly, particularly in the field of public health. As of 11 February 2025, over one million HIV-1 nucleotide sequences were publicly accessible through the Los Alamos HIV Sequence Database (https://www.hiv.lanl.gov, accessed date 11 February 2025). In addition to global databases, many countries maintain national repositories that are accessible exclusively to government agencies responsible for HIV prevention, diagnosis, and treatment. These genomic datasets, when combined with epidemiological information, are now widely used to estimate the rate of HIV-1 spread in specific vulnerable populations and geographic regions, identify transmission hotspots in real time, evaluate the effectiveness of public health interventions, and more. However, the proportion of HIV-infected individuals whose viral sequences have been analyzed varies considerably by region. Developed countries tend to have much higher sequencing coverage compared to developing nations. This disparity also extends to the number of molecular epidemiological studies conducted, which remains significantly lower in developing countries—despite their higher HIV prevalence and larger populations of infected individuals.

As of 31 December 2023, the prevalence of HIV infection in the Russian Federation reached 0.82% of the total population, with 1,194,130 individuals officially registered as living with HIV [1]. Despite the scale of the epidemic, large-scale molecular epidemiological studies aimed at characterizing its dynamics remain extremely limited—primarily due to the relatively small number of available HIV-1 nucleotide sequences. Most published studies to date include fewer than 1000 sequences and tend to focus on localized analyses—examining the characteristics of the epidemic in specific regions [2,3,4,5,6] or population groups [7,8], or concentrating on the distribution of genetic variants [9,10,11,12,13,14,15,16,17,18,19,20]—rather than providing a comprehensive national-level overview.

The HIV/AIDS epidemic in Russia is driven by multiple HIV-1 group M subtypes and remains highly dynamic. Prior to the mid-1990s, the number of HIV infections in the country was relatively low and primarily associated with nosocomial outbreaks caused by HIV-1 subtype G [21], infections among men who have sex with men (MSM) involving subtype B [22], and heterosexual transmission (HET) linked to various African clades [23]. A sharp increase in HIV-1 cases followed the introduction of sub-subtype A6 strains among injecting drug users (IDUs), believed to have entered the country via Ukraine [24,25,26]. Concurrently, a local HIV-1 sub-epidemic emerged in the Northwestern Federal District (including Kaliningrad and Saint Petersburg), driven by the spread of the circulating recombinant form 03_A6B [27,28,29]. Soon thereafter, the co-circulation of sub-subtype A6 and a non-African lineage of 02_AG—initially detected in Uzbekistan [30]—led to the emergence of 63_02A6, which fueled a localized sub-epidemic in Siberia [18,31]. Currently, alongside the dominant sub-subtype A6, other HIV-1 subtypes and circulating recombinant forms (CRFs), including subtypes B, 02_AG, C, G, and others, continue to be identified across different regions of Russia [20,32,33].

In the present study, we utilized HIV-1 nucleotide sequences along with clinical and epidemiological data from Russian citizens living with HIV, as recorded in the national database on HIV drug resistance. The objective of this study was to perform a comprehensive molecular epidemiological analysis across an extended period of the HIV epidemic in the Russian Federation, using the most robust and well-characterized cohort currently available.

## 2. Materials and Methods

### 2.1. Ethics Statement

This study was reviewed and approved by the Ethics Review Committee of the Central Research Institute of Epidemiology (Moscow, Russia) in accordance with relevant local laws and the ethical requirements of the Declaration of Helsinki; the approval number is 152 (3 July 2025). The informed written consent of each HIV-infected patient or the patient’s legal guardian was obtained prior to the sampling and collection of clinical, demographic, and epidemiological data. All the data were anonymized and coded at a national level before being uploaded to the Russian HIV Antiviral Drug Resistance (RuHIV, https://ruhiv.ru/, accessed date 11 February 2025) database.

### 2.2. Overview of Data

This study utilizes data from the RuHIV database, which contains 13,578 unique patient records and 14,397 partial or full HIV-1 *pol* gene sequences obtained from blood samples (plasma or peripheral blood mononuclear cells) using Sanger sequencing. We focused our analysis on partial *pol* gene sequences encompassing the entire protease and part of the reverse transcriptase region. This approach ensured an optimal balance between broad geographic representation and sequence length sufficient to retain informative nucleotide variation. Inclusion criteria for this analysis were: (i) high-quality sequences (sequences passed World Health Organization (WHO) quality control tool) and availability of key metadata, including (ii) country of infection and residence, (iii) year of sampling, (iv) HIV transmission risk category, (v) age at HIV diagnosis, and (vi) sex. Based on these criteria, 9500 patients with HIV-1 diagnosed between 1988 and 2023 from all 8 Federal 135 Districts (FDs) and 78 of Russia’s 89 federal subjects (87.6%) were included; virologic data (genomes) were obtained from 1995 to 2023. For patients with multiple sequences, only the earliest sequence was retained, resulting in one sequence per patient.

### 2.3. Sequence Alignment and Subtype Assignment

Sequences were aligned using the MAFFT v7.0 program [34]; the total length of the alignment was 1116 nucleotides, which covered the entire protease (PR) and partial reverse transcriptase (RT) regions (positions 2253–3351, HXB2-numbering). Subtypes were determined using the HIVdbProgram Sequence Analysis (https://hivdb.stanford.edu, accessed date 11 February 2025), COMET HIV-1 [35] and REGA Subtyping Tool v3.0 [36]. If subtype assignment by the tools was discordant, maximum likelihood (ML) phylogenetic analyses was used. ML phylogenetic trees were inferred with IQ-TREE v2.4.0 [37] with 1000 replicates for bootstrap under the General Time Reversible (GTR) model of nucleotide substitution with proportion of invariable sites (+I) and gamma-distributed rate variation among sites (+G), which was selected as best fitting in jModelTest v2.1.7 [38]. For subtype verification by ML approach, pure subtype and recombinant forms reference sequences for HIV-1 group M, as well as the BLAST v1.4.0 hits, were added to the alignment by Los Alamos HIV Sequence Database. HIV-1 subtype B and 02_AG *pol* sequences were additionally combined with the B-former Soviet Union (FSU; this variant is also known as IDU-B) [29,39,40] (GenBank accession number DQ207943 and JX500708) and the 02_AG_FSU_ [31,41] (GenBank accession number AY829207 and JX500703) reference sequences and were subjected to new rounds of ML phylogenetic analysis for confirmation of clade assignment. After subtype assignment, HIV-1 sequences were grouped into subtype-specific datasets. Alignment gaps and ambiguous bases were not taken into consideration.

### 2.4. Definition of HIV Cluster

In this study we defined two types of HIV clusters: phylogenetic clusters (PCs) and molecular transmission clusters (MTCs). In the first case, cluster was defined as a viral lineage that gives rise to a monophyletic sub-tree of the overall phylogeny with strong statistical support. We used the bootstrapped ML method with Shimodaira–Hasegawa-like approximate likelihood ratio test (SHaLRT), considering clusters with both SHaLRT and bootstrap support values > 0.9 as reliable. In the second case, clusters were defined as groups of cases within a transmission network whose viral sequences are more closely related to each other than to sequences from the broader population. In this network framework, nodes represent individual sequences or patients, while edges denote potential transmission links. Nodes with a higher number of connections (edges) were interpreted as having elevated transmission potential, with nodes possessing ≥4 links flagged as suspected high-transmission-risk cases. The construction of effective transmission network for identifying potential transmission clusters was made using the MicrobeTrace tool [42] with a Tamura–Nei (TN93) nucleotide substitution model and individual optimal genetic distance threshold for each subtype-specific datasets: 0.0075 substitutions/site (0.75%) for subtype A6 and 63_02A6, and 0.015 substitutions/site (1.5%) for subtype B, 02_AG_FSU_, 03_A6B and 14/73_BG; the selection of the optimal threshold (i.e., the distance when the maximum ratio of the number of clusters to distance is detected) was also carried out using MicrobeTrace v0.9.1 (Appendix A). MTCs were then divided into small clusters (3–9 nodes), medium clusters (10–20 nodes) and large clusters (>20 nodes). MTCs containing at least one sequence sampled in the last five years (during 2018–2023) considered active. After constructing the transmission networks, we investigated factors associated with clustering and with having a high number of transmission links (≥4) using multivariable regression analysis. The model included the following covariates: age at HIV diagnosis, sex, transmission risk category, antiretroviral therapy (ART) status, HIV-1 subtype, place of residence (domicile), and sampling year. We also assessed assortativity within transmission clusters and dyads by attributes such as age, sex, transmission risk, and domicile. Assortativity measures the tendency of nodes in a network to connect with other nodes that share similar characteristics. It is quantified using the assortativity coefficient (*r*), which ranges from −1 to 1. A positive r indicates assortative mixing, where similar nodes are more likely to be connected; a negative r indicates disassortative mixing, where dissimilar nodes are more likely to connect; and an r near zero suggests no preferential mixing based on the attribute being analyzed [43].

### 2.5. Phylodynamic and Phylogeographic Reconstructions

The time to the most recent common ancestor (tMRCA) and the ancestral geographic movements of HIV-1 subtypes circulating in Russia were estimated through phylogenetic analyses. For each index sequence, the 500 most similar sequences were retrieved from the NCBI GenBank database (https://www.ncbi.nlm.nih.gov/genbank/, accessed date 11 February 2025) using the BLAST v1.4.0 tool. After removing (i) duplicate sequences from the same individual, (ii) sequences shorter than 948 nucleotides, and (iii) sequences lacking sampling year or country information, we compiled the Russian sequences from our dataset with reference sequences into subtype-specific alignments. These alignments were analyzed using a local instance of the Nextstrain workflow [44], implemented in Python v3.7.5 and utilizing augur v6.1.1. ML trees were constructed using IQ-TREE under the GTR model of nucleotide substitution. Time-resolved phylogenies were generated with TreeTime using a least-squares approach for rooting. To reduce sampling bias associated with overrepresentation from specific locations or time periods, we employed grouped uniform sampling via the --sequences-per-group option in augur, grouping by region, year, and month. The number of sequences per year per region was determined separately for each subtype-specific dataset (Appendix A). Migration events were inferred by counting the number of parent–child branches in the phylogenetic trees where geographic states differed, providing a measure of transmission between regions. The temporal signal of each subtype-specific dataset was assessed using TempEst v1.5 [45] by regressing root-to-tip genetic distances against sampling dates derived from the ML trees (Appendix A).

### 2.6. Statistical Analysis

We report the descriptive statistics of demographic and clinical characteristics of patients with HIV included in this study. Continuous and categorical variables are presented as medians and interquartile ranges (IQR) and numbers and percentages (%), respectively. Baseline characteristics were analyzed using the Mann–Whitney U-test (non-categorical variables) and Pearson chi-square (χ^2^) or Fisher’s exact test (categorical variables). To analyze the predictors of clustering depending on demographic and clinical HIV-infection characteristics, we employed the multivariate logistic regression model via Newton’s method with results reported as odds ratio (OR) and 95% confidence interval (CI). For all tests, *p* < 0.05 was considered statistically significant. Statistical analyses were performed using STATISTICA v.10.0 software (StatSoft, Tulsa, OK, USA).

## 3. Results

### 3.1. Demographic Characteristics

The median age at study entry was 30.0 years (IQR: 24.0–38.0) (Table 1). A substantial proportion of participants (68.4%) were between 20 and 39 years of age. The majority of individuals were male (62.2%). The most common reported transmission route was HET (43.6%), followed by IDUs (26.7%), unspecified sexual transmission (ST: 6.7%), and MSM (5.6%). Overall, HET, MSM, and individuals with unspecified sexual transmission accounted for 55.9% of study participants (Table 1).

### 3.2. HIV Subtype Diversity

In the study sample, HIV-1 subtype A6 was the predominant variant, accounting for 80.6% (95% CI, 79.8–81.4) of infections, followed by 63_02A6 (7.9%; 95% CI, 7.3–8.4), subtype B (5.6%; 95% CI, 5.1–6.1; including the IDU-B variant [40], which made up 9.8% [*n* = 52/531] of all B subtypes), 02_AG_FSU_ (1.2%; 95% CI, 0.1–1.4), 03_A6B (0.7%; 95% CI, 0.6–0.9), and 14/73_BG (0.6%; 95% CI, 0.5–0.8) (Table 1). Other subtypes collectively accounted for 3.4% (95% CI, 3.1–3.8) of cases and included 01_AE, 02_AGAfrican, 06_cpx, 11_cpx, 18_cpx, 19_cpx, 20_BG, 24_BG, 141_BF1, A1, A7, C, D, F1, G, A6B unique recombinant forms (URFs), non-A6B URFs, and genotypes that could not be definitively classified based on partial sequences (designated as like-[subtype]). A detailed description of these subtypes is provided in the captions to Table 1 and Appendix A. Overall, non-A6 HIV-1 infections were more common in males (29.2%, 1644/5636) compared to females (14.3%, 554/3864) (OR, 1.51; 95% CI, 1.35–1.70; *p* < 0.001) (Table 1 and Table 2). Non-A6 subtypes were also significantly more frequent among MSM (57.7%, 309/535; *p* < 0.001), and less frequent among HET (15.9%, 658/4140; *p* < 0.001) and IDUs (13.6%, 346/2541; *p* < 0.001). The odds of a non-A6 infection among MSM were over seven times higher than among HET (OR, 7.33; 95% CI, 5.86–9.17; *p* < 0.001), primarily driven by subtype B (OR, 18.42; 95% CI, 14.42–23.52; *p* < 0.001), which constituted 38.0% (202/531) of infections in the MSM group. Similarly, individuals with “other” risk factors were more likely to have non-A6 subtypes compared to HET (OR, 1.34; 95% CI, 1.24–1.64; *p* < 0.001). Regarding geographical distribution, individuals from the Central Federal District were less likely to be infected with non-A6 subtypes compared to those infected in other regions combined (OR, 1.25; 95% CI, 1.12–1.39; *p* < 0.001) (Table 2).

### 3.3. The Spatial and Temporal Distribution of HIV-1 Subtype

A subgroup analysis by two-year diagnosis intervals revealed substantial fluctuations in the overall prevalence of HIV-1 subtypes over time (Figure 1). Subtype A6 remained dominant throughout the study period, peaking in 2001–2002 with a prevalence of 94.9%, followed by subtype B (2.3%), “other” subtypes, 03_A6B, and URF_A6B (each with ~0.7%). However, the prevalence of A6 declined steadily thereafter, reaching 61.4% in 2021–2022 (OR: 11.67; 95% CI: 7.76–17.55; *p* < 0.001). In contrast, the proportions of 63_02A6, subtype B, 02_AG_FSU_, and 14/73_BG increased over time. Specifically, 63_02A6 rose from 0.4% to 23.9% (OR: 73.52; 95% CI: 18.03–299.83; *p* < 0.001). Subtype B also exhibited a consistent upward trend from 2.3% in 2002 to 7.6% in 2020 (OR: 3.41; 95% CI: 1.87–6.21; *p* < 0.001), before stabilizing and slightly decreasing to 4.4% in 2023. Similarly, 02_AG_FSU_ increased from 0.2% in 2002 to 3.1% in 2023 (OR: 19.41; 95% CI: 2.54–148.19; *p* < 0.001). 14/73_BG displayed a transient spike in 2017–2018 (1.4%) followed by a decrease to 0.4% in 2023. The share of “other” subtypes rose from 0.6% in 2001–2002 to 6.6% in 2021–2022. 03_A6B prevalence remained relatively stable, aside from fluctuations in the early years of the epidemic (Figure 1). Overall, the prevalence of non-A6 subtypes grew by 33.5 percentage points over the past two decades—from 5.1% in 2002 to 38.6% in 2023 (OR: 11.67; 95% CI: 7.76–17.55; *p* < 0.001). As with temporal trends, the geographical distribution of HIV-1 subtypes varied across the FDs of Russia (Figure 2). From 1987 to 2023, the overall prevalence of non-A6 subtypes ranged from 5.9% in the Volga FD to 38.3% in the Siberian FD. In addition to Siberia, the Far Eastern and North Caucasian FDs also exhibited high proportions of non-A6 subtypes (23.9% each). In the North Caucasian (20.5%; OR: 3.72; 95% CI: 2.13–6.52; *p* < 0.001) and Siberian (34.1%; OR: 18.45; 95% CI: 12.73–26.75; *p* < 0.001) FDs, this increase was mainly driven by the rise in 63_02A6. By contrast, in the Central (6.3%; OR: 1.65; 95% CI: 1.34–2.02; *p* < 0.001), Northwestern (11.0%; OR: 4.52; 95% CI: 2.83–7.20; *p* < 0.001), and Volga (2.3%; OR: 1.94; 95% CI: 0.99–3.82; *p* = 0.049) FDs, subtype B was the main contributor to the rise in non-A6 prevalence.

Notably, in the Ural FD, the increase in non-A6 cases was primarily attributed to 03_A6B (5.8%; OR: 1.75; 95% CI: 1.04–2.92; *p* = 0.043). URF_A6B had the highest share in the Central FD (1.4%). The highest prevalence of 02_AG_FSU_ was observed in the Central (1.7%) and North Caucasian (1.4%) FDs. 14/73_BG was most prevalent in the Volga, North Caucasian, and Southern FDs (approximately 1% in each). The proportion of “other” subtypes remained relatively stable across regions (approximately 3%), except for the Ural (1.0%) and Volga (0.4%) FDs, where significantly lower proportions were observed.

### 3.4. Molecular Network Analysis and Transmission Clusters

A total of 9175 individuals infected with six major HIV-1 subtypes were included in the analysis of cluster membership correlates. Of these, 1844 participants (20.1%) were part of MTCs, while the remaining 7331 (79.9%) were singletons (Appendix A). In total, 421 MTCs were identified, with cluster sizes ranging from 2 to 394 participants. Nearly half of the clustered individuals (*n* = 893; 48.4%) were part of 256 active clusters, defined as containing at least one sequence sampled between 2018 and 2023. Among all clustered participants, 638 (34.6%) were in dyads (*n* = 319), 380 (20.6%) were in small clusters (*n* = 92), and 107 (5.8%) and 719 (39.0%) were in medium (*n* = 7) and large clusters (*n* = 3), respectively (Figure 3, Appendix A). Notably, the emergence of dyads and small MTCs became more prominent from 2002 onward, whereas the proportion of individuals in large clusters declined steadily over time (Appendix A). Subtype A6 formed the largest number of clusters and had the highest total number of clustered individuals, with 325 clusters encompassing 1223 individuals (16.0% of 7659 A6 cases). HET was the predominant route (45.1%, *n* = 551), followed by IDUs (31.2%, *n* = 382). Among the 747 individuals infected with 63_02A6, 384 (51.4%) were included in 24 clusters, with HET (37.8%, *n* = 145) and IDUs (30.7%, *n* = 118) being the main transmission routes. Subtype B accounted for 45 clusters involving 151 individuals (28.4% of 531 subtype B cases), with MSM as the primary route (39.7%, *n* = 60), followed by HET (33.1%, *n* = 50).

Subtype B accounted for 45 clusters involving 151 individuals (28.4% of 531 subtype B cases), with MSM as the primary route (39.7%, *n* = 60), followed by HET (33.1%, *n* = 50). Among the 111 individuals with 02_AG_FSU_ infection, 35 (31.5%) were clustered in 14 MTCs, primarily via HET (37.1%, *n* = 13) and IDUs (25.7%, *n* = 9). For 03_A6B, 25 of 68 individuals (16.0%) formed three clusters, with HET (44.0%, *n* = 11) and IDUs (20.1%, *n* = 5) as main routes. Among the 59 14/73_BG-infected individuals, 26 (44.1%) formed seven clusters, predominantly transmitted via HET (46.1%, *n* = 12) and MSM (33.8%, *n* = 8). Of the 421 MTCs, the majority were predominantly HET (*n* = 135; 32.1%), followed by IDUs (*n* = 30; 7.1%), MSM (*n* = 28; 6.6%), MTCT (*n* = 5; 1.2%), and nosocomial (NSC) transmission (*n* = 2; 0.5%). Mixed clusters included HET+IDUs (*n* = 71; 16.9%), HET+MSM (*n* = 26; 6.2%), and HET+MTCT (*n* = 11; 2.6%), while other mixed types (e.g., IDUs+MSM, HET+NSC, IDUs+NSC, MSM+NSC) represented less than 1% (Appendix A). A substantial proportion of clusters could not be classified by predominant transmission route: 88 of 327 subtype A6 clusters (26.9%) and 32 of 94 non-A6 clusters (34.0%) (Appendix A). For A6 clusters, HET predominance was most common (*n* = 136; 41.6%), followed by HET+IDUs (*n* = 63; 19.3%) and IDUs alone (*n* = 25; 7.6%). MSM and MSM+HET clusters were more common among non-A6 subtypes (MSM: 9.6%; MSM+HET: 20.1%) than among A6 clusters (MSM: 5.8%; MSM+HET: 2.1%), especially for subtype B (MSM: 15.2%, *p* = 0.023 vs. A6; MSM+HET: 30.4%, *p* < 0.001 vs. A6) (Appendix A). Overall, individuals with subtype A6 were significantly less likely to be part of an MTCs than those with non-A6 subtypes (16.0% vs. 41.0%, *p* < 0.001), with the highest clustering frequency observed among 63_02A6-infected individuals (51.4%). Three large MTCs were identified: one A6 cluster comprising 394 individuals (MTC-0l394), and two 63_02A6 clusters with 238 (MTC-14l238) and 87 individuals (MTC-0l87), respectively.

The large A6 cluster was predominantly male (*n* = 214; 54.3%) and IDU-driven (*n* = 186; 47.2%), and included individuals with the youngest median age at diagnosis (24.0 years; IQR: 20.0–30.0). In contrast, the largest 63_02A6 cluster was HET-dominated (*n* = 84; 35.3%), had an older median age at diagnosis (35.0 years; IQR: 27.0–41.0), and a higher male proportion (*n* = 150; 63.0%). Notably, these large clusters lacked assortativity by transmission category (Figure 4). Overall, high assortativity by transmission route (r > 0.2) was rare and primarily observed in dyads, indicating transmission route concordance was mainly confined to pairs. In contrast, region-based assortativity was consistently high across all cluster sizes (r > 0.35). Age assortativity was also strong in dyads (r > 0.35) across all subtypes. Assortativity by sex varied: it was disassortative in medium-sized subtype B clusters but not in dyads or small clusters; similarly, disassortativity was observed in small clusters of 02_AG_FSU_ and 63_02A6, but not in their dyads or medium clusters. Remarkably, 02_AG_FSU_ was the only subtype with strong assortativity across age, sex, and region for all cluster size categories.

### 3.5. Correlates of Clustering

Among individuals with sequences included in MTCs, 60.2% were male, and 47.0% were aged 30–49 years, with a median age of 36.0 years (IQR, 33.0–40.0). In terms of transmission risk, 42.4% were associated with HET, 28.5% with IDUs, 7.4% MSM, and 4.2% with other sources (MTCT and NSC). Geographically, 74.9% (*n* = 649/866) of clustered individuals resided in European Russia, across five federal districts (Central, North Caucasian, Northwestern, Southern, and Volga), with the highest proportion from the Central Federal District (43.5%) (Table 3). Multivariable analysis revealed several factors significantly associated with increased odds of clustering. These included being younger than 30 years (OR, 1.31; 95% CI, 1.03–1.66), not receiving antiretroviral therapy (OR, 2.70; 95% CI, 2.40–3.04), IDUs as a transmission route (OR, 1.28; 95% CI, 1.01–1.62), and NSC transmission (OR, 5.26; 95% CI, 2.83–9.76) (Table 3). Notably, being part of an MSM cohort did not significantly increase the likelihood of clustering (OR, 1.11; 95% CI, 0.86–1.44; *p* = 0.410). Clustering was significantly more frequent among individuals infected with non-A6 subtypes, particularly 63_02A6 (OR, 5.70; 95% CI, 4.82–6.73). Regional differences were also observed: individuals diagnosed in the North Caucasian Federal District had significantly higher odds of clustering (OR, 1.81; 95% CI, 1.35–2.43), while those in the Northwestern Federal District had significantly lower odds (OR, 0.28; 95% CI, 0.20–0.39). Further analysis comparing individuals with high linkage (≥4 links) to those in low-linkage clusters revealed distinct patterns (Appendix A).

Higher odds of being in highly linked networks were observed among those infected with 63_02A6 (OR, 7.36; 95% CI, 5.57–9.71) or 03_A6B (OR, 3.66; 95% CI, 1.59–8.40), individuals with unknown transmission risk (OR, 1.60; 95% CI, 1.11–2.30), those reporting sexual risk without specification (OR, 2.17; 95% CI, 1.27–3.71), and individuals with an IDUs transmission history (OR, 3.05; 95% CI, 2.31–4.03). Conversely, individuals infected with subtype B (OR, 0.58; 95% CI, 0.36–0.93) and those residing in the Northwestern Federal District (OR, 0.13; 95% CI, 0.04–0.42) were significantly less likely to belong to highly linked networks. Additionally, a temporal trend was observed, with individuals in high-linkage networks more likely to have been sampled in earlier years (Appendix A).

### 3.6. Origin and Migration Pathway

Following the identification of MTCs and molecular network analyses, we conducted discrete-geographic phylodynamic analysis to estimate viral migration events involving Russia, including both international introductions and within-country movements. Consistent with previous approaches, our analysis focused on six major HIV subtypes (Figure 5; Appendix A). Of all migration events involving Russian sequences (*n* = 4091), 14.7% (*n* = 603) were international in origin. Among these, 54.9% (*n* = 303) were from Ukraine, and FSU countries collectively accounted for 83.8% (*n* = 501) of international viral introductions. This strong bias toward Ukraine and other FSU countries was largely due to the disproportionate presence of subtype A6, which alone accounted for 425 out of 603 international introductions. Among subtypes, the highest proportion of international introductions was observed for 02_AG_FSU_ (44.1%), with 65.0% originating from Uzbekistan. In contrast, subtype 63_02A6 had the fewest international introductions (0.1%), all from FSU countries. Conversely, international introductions of subtype B (10.3%) and 14/73_BG (6.7%) were predominantly from non-FSU countries—74.1% and 100%, respectively. Within Russia, the majority of viral dispersal originated from the Moscow capital region (the city of Moscow and Moscow Oblast), accounting for 78.7% of within-country transmission events, followed by Tver Oblast (66.1%) (Appendix A). Moreover, Moscow and/or Moscow Oblast were inferred as the ancestral location for at least 14 phylogenetic clusters or sub-clusters, significantly more than any other region. Notably, the capital region contributed minimally to the spread of subtype 63_02A6, which remains primarily disseminated within the Siberian Federal District. A similar pattern was observed for 03_A6B, whose spread is concentrated in the Northwestern (via Kaliningrad Oblast) and Ural (via Sverdlovsk Oblast) federal districts (Figure 5). Overall, approximately half of all identified phylogenetic clusters (PCs), encompassing 13.8% (1375/9979) of Russian sequences, were of Russian origin (Table 4). Subtype A6 was represented by six PCs, including the largest (Cluster #1), with 540 infected individuals, 76.1% of whom were Russian. Most PCs were geographically mixed, with no single federal subject contributing more than 35% of sequences. Only two PCs were regionally concentrated: Cluster #3 (78% from Krasnoyarsk Krai) and Cluster #5 (98% from Orel Oblast). Ukraine was inferred as the ancestral location for over half of all A6 clusters. Molecular clock dating indicated that A6 clusters originated between 1996 (Cluster #1) and 2004 (Cluster #5), with the estimated tMRCA for the overall A6 dataset being 1993.2. For subtype B, most PCs were also non-Russian in origin, primarily from Europe or the Americas, rather than from FSU countries. Similar to A6, most subtype B PCs were mixed geographically. The exceptions were two sub-clusters with regional predominance: Cluster #1.3 (70% from Krasnodar Krai) and Cluster #2.1 (60% from Primorsky Krai). Cluster #2 was nested among basal sequences from Poland, suggesting a likely introduction from Poland into the FSU.

By contrast, the Primorsky sub-cluster likely resulted from a secondary dissemination within Russia. The earliest tMRCA for subtype B was estimated at 1978.1 (Cluster #1), with most clusters dating between 1982.7 and 1988.7. The 63_02A6 showed entirely Russian origin for all its PCs, as expected (Table 4). Clusters #1, #3, and #5 circulated primarily in the Siberian Federal District, comprising 50.0% of all 63_02A6 Russian sequences. Cluster #2 was dominant in the Central FD (Orel Oblast), accounting for 90% of 63_02A6 cases from that region. Cluster #4 circulated mostly in the Russian Far East, comprising about 80% of local 63_02A6 cases. Almost all 63_02A6 cases from the North Caucasian and Southern FDs fell into sub-cluster #1.1, likely representing multiple introductions from Novosibirsk Oblast or, at a minimum, strong epidemiological links to it. The earliest tMRCA for this subtype was estimated at 2004.3, with most clusters dating between 2008.8 and 2012.2. All Russian cases of subtype 02_AG_FSU_ were distributed across five clusters. The largest proportion (23.0%) was found in a mixed-origin Cluster #1, the only one inferred to have originated in Russia, likely due to secondary dissemination within the Moscow capital region. Another notable cluster (#3), conditionally regional, was centered in Lipetsk and likely originated from Central Asia. Both clusters had similar estimated tMRCA dates around 2005.1–2006.0. Analysis of 03_A6B infections identified two mixed-origin PCs, with one large cluster accounting for 96.2% of all Russian sequences. Both originated in Kaliningrad Oblast. Cluster #1 was more recent, with a tMRCA of 1993.4, while Cluster #2 dated to approximately 1996.3. Finally, all cases of 14/73_BG were concentrated in two clusters, with ongoing dissemination from their root location in Spain (Table 4). Cluster #1 included 95%.0 of Russian samples and contained two sub-clusters reflecting secondary dissemination: one from Moscow to the surrounding capital region (#1.1), and another to the Republic of Tatarstan (#1.2). The tMRCA for these sub-clusters was estimated at 2003.3 and 2008.0, respectively, with the cluster as a whole dated to 1999.9.

## 4. Discussion

In this study, we employed a comprehensive statistical and phylogenetic approach to analyze a large national HIV-1 dataset comprising over 9500 individuals diagnosed across a 30-year period. This extensive analysis offers novel insights into the highly dynamic HIV-1 epidemic in Russia, which has evolved from being dominated by subtype A6 to encompassing a more diverse mix of subtypes and CRFs. Our results demonstrate that this trend toward diversification has continued in recent years, coinciding with a shift in the predominant mode of transmission—from injection drug use to sexual transmission. Notably, this study is the first to explore HIV-1 molecular transmission networks on a national scale in Russia by integrating molecular data with epidemiological, demographic, and behavioral information. This enabled us to examine local transmission dynamics and reveal cooperative transmission networks between key populations. In addition, our analysis sheds light on how the Russian HIV-1 epidemic is interconnected with global epidemics, highlighting the impact of both international viral introductions and domestic viral migration. These findings underscore the significant role of migration—both cross-border and internal—in shaping the country’s HIV landscape.

Despite the growing diversity, subtype A6 remains the most prevalent variant, accounting for 80.9% of cases in the cohort. However, non-A6 infections represent a substantial proportion of the epidemic, with 63_02A6 (7.9%) being the most common among them, followed by subtype B (5.6%), 02_AG_FSU_ (1.2%), and 03_A6B (0.7%). Regional variations in subtype prevalence were observed: 63_02A6 was more frequent in the Siberian (34.1%) and North Caucasian (20.5%) FDs, subtype B was predominant in the Northwestern FD (11.0%), 03_A6B in the Ural FD (5.3%), and URFs/others in the Central FD (4.4%). These findings are broadly consistent with previous reports from across Russia [15,47,48,49]. A notable finding was the increasing detection of subtype 14/73_BG. In our previous studies conducted in 2015 and 2019 [13,32], this subtype accounted for less than 0.01% of infections, whereas it now comprises 0.6%—indicating ongoing spread. Additionally, at least 10 other minor “pure” subtypes, CRFs, and unique recombinant forms (URFs) were identified, emphasizing the growing genetic heterogeneity of HIV-1 in Russia. One of the most striking trends was the marked decline in subtype A6 prevalence over the past decade (−33.5%), paralleled by significant increases in 63_02A6 (+23.5%), subtype B (+2.1%), and other subtypes/CRFs (+6.0%). The rapid rise of 63_02A6 is particularly significant and suggests it may have biological advantages over other subtypes in certain high-risk populations. Although the underlying molecular mechanisms remain to be elucidated, previous studies [18,50] have proposed that 63_02A6 may possess higher replication capacity and enhanced viral fitness, potentially contributing to its rapid dissemination. The increasing share of non-A6 subtypes appears to correlate with the growing proportion of infections acquired through sexual transmission. This shift likely reflects broader epidemiological changes, including international travel, migration-related introductions, and enhanced internal transmission of previously imported subtypes. Together, these trends suggest that the epidemic in Russia is undergoing significant transformation and may become even more genetically and epidemiologically complex in the future.

As the most dominant HIV-1 strain in Russia, subtype A6 has played a central role in shaping the country’s epidemic. Our analysis revealed that the majority (74.2%) of subtype A6 infections were associated with HET and IDUs, while only 3.0% occurred in MSM. HET individuals had at least seven times greater odds of acquiring an A6 infection than MSM with non-A6 subtypes, and the odds of A6 infection were nearly equal between HET and IDUs groups. We also observed a higher proportion and odds of A6 infection among females compared to males, which aligns with the lower proportion of MSM (a predominantly male group) among A6 cases. Specifically, the proportion of MSM among individuals infected with non-A6 subtypes was about five times higher than among those with A6. These findings are consistent with previous research [20,32,49], confirming that subtype A6 remains the predominant HIV-1 variant among HET and IDUs populations in Russia. Importantly, our results suggest a shift in the transmission dynamics of subtype A6. While IDUs were historically the primary transmission group, heterosexual contact has now surpassed IDUs as the leading transmission route for A6 infections nationwide (45.5% vs. 28.7%). In contrast, subtype B and 14/73_BG infections in Russia are predominantly found in the MSM population, accounting collectively for 41.9% of cases. These subtypes are much less prevalent among IDUs (1.6%), HET individuals (4.6%), and children born to HIV-positive mothers. Although the first confirmed HIV-1 case of 14/73_BG in Russia (diagnosed in 2005) occurred in a IDUs [32], subsequent cases have been mainly linked to heterosexual and MSM transmission, as shown in our study. It is worth noting that while certain HIV-1 subtypes initially appeared to be concentrated within specific high-risk groups (e.g., subtype B among MSM or subtype A6 among IDUs), these distinctions are gradually becoming less pronounced due to ongoing viral mixing between transmission networks. For example, the detection of subtype B—traditionally associated with MSM—among IDUs and heterosexual women suggests increasing transmission between risk groups and the general population. Conversely, the rising prevalence of non-B subtypes among MSM, as supported by at least one other study [7], points to a broader diversification of HIV-1 strains within this group. According to our findings, non-B subtypes have surpassed subtype B in cumulative HIV cases among MSM (42.2% vs. 37.7%), with subtype A6 now being the predominant variant. Additionally, we report that the B_FSU variant, once concentrated mainly among IDUs, is now found across other transmission groups, including HET and MSM. Similar shifts in the distribution of this subtype—from IDUs to other groups—have also been reported in Ukraine [51], further highlighting the dynamic nature of HIV-1 transmission patterns in the region.

We identified that one-fifth of the study population was linked through closely related MTCs. More than 60% of these clusters appeared to be potentially active, representing either newly detected clusters or ongoing expansions of existing ones. Based on the dates of HIV diagnosis among individuals in MTCs, the growth of older (established) clusters appears to contribute as significantly to the ongoing Russian epidemic as newly emerging clusters. Notably, the high proportion of young individuals in MTCs who do not belong to traditional high-risk groups highlights the growing vulnerability of the general population to HIV-1 transmission. Clustering was associated with several demographic and behavioral factors, including being male, under the age of 30, and reporting IDUs or HET. Clustering was also more frequently associated with non-A6 subtypes, suggesting that these individuals are disproportionately involved in transmission networks. These same factors were observed in individuals with multiple potential transmission links (i.e., >4), who may act as “super-spreaders” and should be prioritized for targeted public health interventions [52]. While MSM were less likely to be found in clusters at the time of analysis, they have the potential to form super-spreader links, particularly in contexts such as chemsex [53,54]. Interestingly, although ART was negatively associated with MTCs membership overall, a notable proportion of individuals in clusters (13.2%) were on ART. This may be due to several reasons: transmission may have occurred before the initiation of ART or during early treatment, when viral replication remained unsuppressed; alternatively, treatment failure due to poor adherence or the emergence of drug resistance mutations could have facilitated transmission.

We also found subtype-specific differences in clustering patterns. As expected, the subtype A6 network exhibited the highest number and largest sizes of clusters, including the largest MTCs, which comprised 394 individuals. Despite this, the clustering rate within the subtype A6 population was the lowest (16.0%), suggesting that although the subtype is widely spread, its network is more diffuse. Nevertheless, subtype A6 continues to play a major role in driving the national HIV-1 epidemic in Russia. The 63_02A6 network showed the second-highest clustering rate but involved generally smaller cluster sizes, with the exception of two large MTCs consisting of 80 and 253 members. In contrast, subtype B, 02_AG_FSU_, and 14/73_BG networks were mainly composed of small MTCs, indicating more localized or fragmented epidemic patterns. The more limited clustering of these subtypes can partly be explained by their initial emergence through multiple independent introductions, as discussed later in the text. We also found evidence of regional assortative mixing within transmission clusters—i.e., most links involved individuals from the same geographical region, with a few exceptions. This suggests that local transmission, rather than frequent cross-regional spread, is the main driver of the epidemic. Such correlations between specific regions and subtypes could support the design of geographically targeted intervention strategies to address areas with increasing transmission intensity. With respect to transmission routes and risk populations, networks associated with subtypes A6, 63_02A6, 03_A6B, and partly 02_AG_FSU_ shared common features: they were primarily driven by IDUs or HET, involved mostly individuals under 35 years of age, and exhibited a male-to-female ratio of less than 2:1 (with 03_A6B showing a female predominance). These findings suggest that both heterosexual and IDUs routes remain the primary modes of transmission for these subtypes and involve both men and women. A notable finding was the frequent linkage between IDUs and heterosexual women, suggesting that many IDUs may act as a “bridging” population, facilitating transmission to women. Some subtype B and 14/73_BG MTCs consisted entirely of men, with MSM outnumbering heterosexuals. This pattern may reflect misreporting of transmission risk, as stigma and discrimination could lead many MSM to underreport or conceal their sexual orientation [55,56,57]. Another plausible explanation is that some men identifying as heterosexual may engage in bisexual behavior and serve as additional bridging links between MSM networks and the general population—similar to the role IDUs play in heterosexual transmission. In contrast, networks involving subtype B and 14/73_BG were primarily composed of MSM and, to a lesser extent, heterosexual individuals. These networks exhibited a male-to-female ratio of over 6:1, underscoring the predominant role of MSM in driving their spread. However, risk behaviors were often mixed within MTCs. In fact, only 35% of MTCs could be clearly classified by a single transmission risk group (HET, IDUs, or MSM). The three largest MTCs included individuals from all transmission categories, though HET and IDUs were most frequently represented. Overall, mixed-risk transmission clusters are likely to be critical drivers of ongoing HIV-1 spread in Russia. Identifying and intervening within these bridging populations—who facilitate cross-group transmission—is essential for preventing further spread into the broader population.

Our phylogeographic analyses demonstrated a significant association between all analyzed HIV-1 subtypes and various foreign populations: Eastern Europe and Central Asia (subtypes A6, 63_02A6, 02_AG_FSU_, and 03_A6B) and Western Europe and America (subtype B and 14/73_BG). These associations align with the known geographical distributions of these variants [18,33,58,59,60,61,62,63,64,65,66,67]. In the present study, we found that local phylogenetic clusters of the most prevalent subtypes originated between the late 1970s (subtype B) and the mid-2000s (63_02A6). These dates should be considered the latest boundary for the onset of the HIV-1 epidemic caused by these variants. Analysis of A6 infections showed that most sequences isolated from Russia formed a few monophyletic clusters, including a large outbreak clade comprising 540 individuals, predominantly among IDUs and HET. The root of this clade was traced to Ukraine. We inferred the introduction of subtype A6 into Russia around 1994, consistent with previous estimates [10]. Surveillance data [24,25] also confirm that the expansion of the A6 epidemic in Russia began in the late 1990s, suggesting that the origin of the locally expanding lineages dates to the mid-1990s, as our analysis supports. These findings collectively indicate that Ukraine served as the initial point for subtype A6 expansion into Russia, followed primarily by local onward transmission, contributing to the HIV-1 spread across most Russian regions, as previously reported [6,68,69,70,71]. While local transmission is the dominant mode of A6 dissemination within Russia, secondary introductions from international sources continue to occur. Notably, 88.3% of these cross-border transmissions originated from FSU countries—primarily Ukraine, Belarus, Uzbekistan, Tajikistan, and Kyrgyzstan. Given their shared Soviet past, these historic ties are unsurprising. Russia, as the most economically developed country in the FSU, attracts labor migrants—mostly men—from neighboring states [72,73,74]. Additionally, tourist flows to culturally rich regions such as the North Caucasus and the Black Sea coast contribute to ongoing cross-border contacts. These factors may help explain the secondary migration events of subtype A6 identified in this study. We also detected limited cross-border A6 transmissions involving non-FSU countries. For example, our results suggest migration pathways between Russia and Bulgaria, Turkey, and Italy, countries that have previously reported A6 transmission involving heterosexual and homosexual contact with individuals from Russia and Ukraine [75,76,77]. Such transmissions may have contributed to the introduction of A6 into Western and Central Europe. As part of the 02_AG_FSU_ sub-epidemic, we inferred several clusters, including one relatively large cluster of 25 members dated to 2005. The ancestral origin of these clades was traced with high probability to Central Asian FSU countries, primarily Uzbekistan and Kyrgyzstan. This supports earlier findings and highlights the central role of these countries in the spread of 02_AG_FSU_ into Russia after 2000 [9,41,65,78,79,80]. Specifically, the vast majority (>90%) of 02_AG_FSU_ introductions into Russia were linked to Uzbekistan and Kyrgyzstan. In contrast, we observed only marginal cross-border 02_AG_FSU_ transmissions from Russia back to FSU countries. Although these patterns largely reflect IDUs interactions between Russia and Central Asia, labor migration likely also plays a role. One study reports the dissemination of 02_AG_FSU_ from Uzbekistan to Russia through migrants who acquired HIV via local female sex workers or possibly their spouses [80]. These findings underscore the need for HIV prevention and education programs targeting migrant populations. Regarding subtype B, our analysis revealed multiple introductions into Russia from at least five different countries, followed by local spread among Russians. Large phylogenetic clusters were observed, including groups of 139, 75, and 56 Russian individuals. These clusters originated between 1978 (Western B) and 1989 (B-FSU), indicating a later emergence of B-FSU compared to Western B. This observation is consistent with previous findings based on smaller datasets [70] and other studies [17]. Taken together, the results suggest that subtype B was already circulating among Russians before the A6 epidemic began and was likely the first subtype to establish an epidemic in Russia, albeit initially limited to MSM. As expected, our analysis showed subtype B-FSU was predominantly associated with IDUs, whereas Western B was more common among MSM and HET. However, some intermixing of sequences across key populations was evident, indicating “bridging” of infections from high-risk groups into the general population. As for the migration pathways of subtype B, most Western B infections appeared to originate from Europe and the Americas, likely associated with economic travel. This may explain the presence of HIV-1 sequences from the USA, Italy, the UK, Poland, and other countries that were ancestral to Russian clusters. It is also possible that tourism or immigration to Russia contributed to the introduction of subtype B, particularly through MSM-related transmissions. Our analysis further revealed that the emergence of 03_A6B and 63_02A6 occurred in 1995 and 2005, respectively, with the most probable origins being Kaliningrad and Novosibirsk. These CRFs subsequently spread to other Russian regions and to Armenia, Azerbaijan, Belarus, Lithuania, Kyrgyzstan, Tajikistan, and Uzbekistan. This supports the hypothesis that Russia serves as a central hub for the dissemination of these CRFs within the FSU countries. A more detailed analysis of 03_A6B and 63_02A6 is presented elsewhere [14,18]. Lastly, we report on the dissemination of 14/73_BG in Russia. Our data suggest this epidemic originated from a single viral clade, dated to around 2000, with a probable origin in Spain. This is somewhat unexpected given that most 14/73_BG infections in Russia are found among MSM and HET. One might anticipate a pattern similar to that of subtype B, with multiple introductions from various countries leading to smaller sub-clusters. Earlier studies proposed that 14/73_BG originated in the Iberian Peninsula [59,61] and has been rising in prevalence in some Eastern European countries, including Romania [81] and Greece [82], with origins linked to southwestern Europe. However, we found no associations between strains from these countries and Russian 14/73_BG infections. This could be due to the limited number of sequences available from those regions at the time of our study and the broader under-sampling relative to the epidemic in Spain. Assuming sufficient sampling, the presence of only Russian sequences within this single large clade supports the hypothesis of local onward transmission of 14/73_BG after a single introduction from Spain. Our data do not support multiple importation events as the primary driver of this sub-epidemic in Russia.

This study has several limitations that should be considered. First, although our sample size was substantial, it represented only approximately 1% of the officially reported HIV-1 cases in Russia [1]. Therefore, the dataset may not be fully representative of the overall population living with HIV in the country. In particular, the overwhelming majority of patients included were Russian citizens, and the proportion of migrants was very small. As a result, the contribution of migrants to MTCs may be underestimated, and unsampled individuals could limit the accuracy of transmission dynamics inferred from the data. Second, the relatively limited sampling depth may have restricted our ability to detect transmission clusters, particularly smaller or emerging ones. At the population level, the actual network of HIV infections may be broader than what was captured in our study, with unrecognized or unsampled transmissions occurring. Third, the absence of data on transmission routes for a substantial portion of the samples, as well as potential misclassification of reported risk factors, could further obscure the true dynamics of transmission. Social stigma and discrimination may discourage individuals from disclosing high-risk behaviors, such as injection drug use, sex work, or same-sex sexual contact. This underreporting may lead to the omission of key individuals from the analysis and hinder accurate characterization of transmission clusters at the population level. Lastly, subtyping analyses in this study were based solely on a fragment of the *pol* gene. This limitation precludes detailed analysis of recombination patterns and may lead to underestimation of the true diversity and prevalence of recombinant forms. Full-genome HIV-1 sequencing would likely improve the detection of recombinant forms, enable confirmation of recombination breakpoints, and provide more accurate reconstructions of transmission histories. Therefore, the findings reported here should be interpreted with appropriate caution.

## 5. Conclusions

This study demonstrates that molecular epidemiological surveillance of HIV-1 in Russia can provide valuable real-time insights into transmission dynamics, especially among high-risk populations. Molecular data can support contact tracing, detect new introductions, and identify growing transmission clusters. Our findings reveal a highly complex and evolving regional epidemic, marked by strong epidemiological links to other countries of the FSU as well as global migration hubs. Subtype A6 remains the most prevalent HIV-1 variant in Russia; however, we also observed a significant increase in the prevalence of non-A6 subtypes, including various CRFs. The ability of these non-A6 variants to form active MTCs indicates that they are now well established in the local population. These results have clear public health implications. They highlight the need to reinforce prevention strategies, particularly those targeting high-risk behaviors and key populations. Continued molecular surveillance will enable the timely detection of new transmission clusters and novel variants, which is essential for designing effective, evidence-based interventions to reduce transmission intensity and improve epidemic control.

## Figures and Tables

**Figure 1 pathogens-14-00738-f001:**
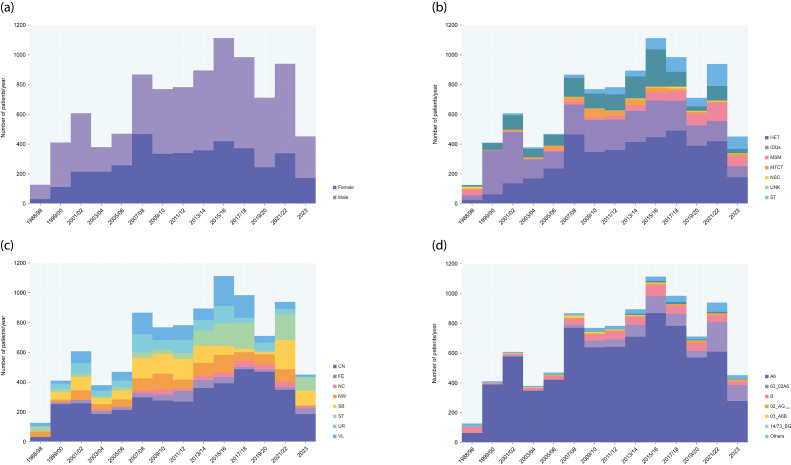
Temporal trends in HIV-1 diagnoses in Russia (1987–2023) by key characteristics. Temporal distribution of sequence-confirmed HIV-1 cases in Russia by: (**a**) Sex; (**b**) Transmission risk group; (**c**) Federal Districts (FDs) of sampling; (**d**) HIV-1 subtype. The x-axis represents the midpoint of each 2-year interval in which patients were diagnosed. HET, heterosexual contact; IDUs, injecting drug users; MSM, men who have sex with men; MTCT, mother-to-child transmission; NSC, nosocomial transmission; UNK, unknown; ST, sexual transmission (unspecified).

**Figure 2 pathogens-14-00738-f002:**
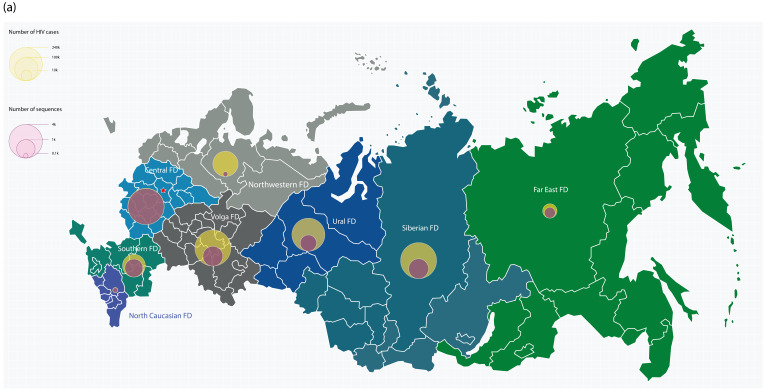
Geographic distribution of HIV-1 subtypes in Russia. (**a**) Map of Russia`s Federal Districts (FDs). Cumulative number of registered HIV cases and analyzed sequences are displayed as color-coded bubbles. (**b**) Map of Federal Districts and sub-regions, color-coded by the number of sequences analyzed. (**c**) Proportional distribution of HIV-1 subtypes within each FD. CN, Central FD; FE, Far Eastern FD; NC, North Caucasian FD; NW, Northwestern FD; SB, Siberian FD; ST, Southern FD; UR, Ural FD; VL, Volga FD.

**Figure 3 pathogens-14-00738-f003:**
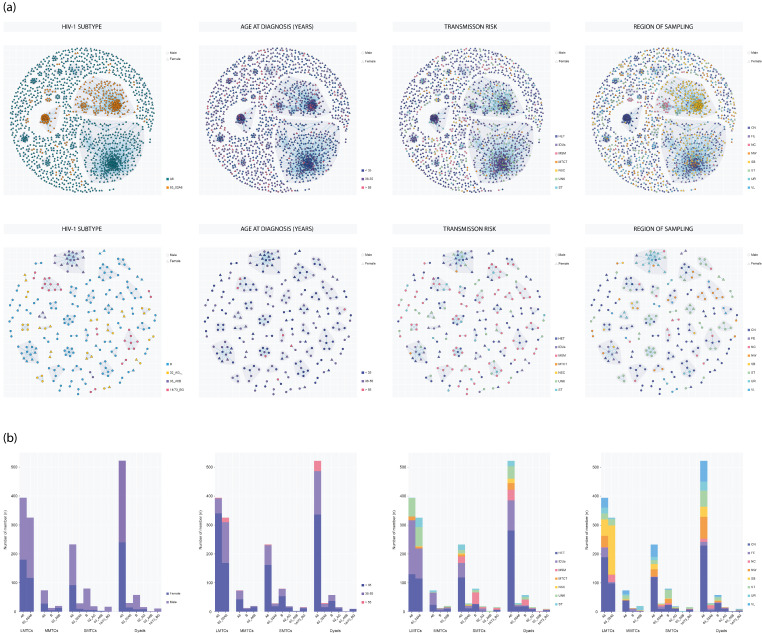
Molecular transmission clusters (MTCs) among individuals with HIV-1 in Russia. (**a**) Visualization of selected MTCs with node attributes showing subtype, age at diagnosis, transmission risk, region (FD), and sex. Each node represents a person; edges indicate pairwise genetic distances ≤0.75% for A6 and 63_02A6 (**top panel**), and ≤1.5% for subtype B, 02_AG_FSU_, CRF03_A6B, and 14/73_BG (**bottom panel**). (**b**) Clustering characteristics of four types of clusters involving a major HIV-1 subtypes in the network. HET, heterosexual contact; IDUs, injecting drug users; MSM, men who have sex with men; MTCT, mother-to-child transmission; NSC, nosocomial transmission; UNK, unknown; ST, sexual transmission (unspecified); CN, Central FD; FE, Far Eastern FD; NC, North Caucasian FD; NW, Northwestern FD; SB, Siberian FD; ST, Southern FD; UR, Ural FD; VL, Volga FD; LMTCs, large MTCs; MMTCs, medium MTCs; SMTCs, small MTCs.

**Figure 4 pathogens-14-00738-f004:**
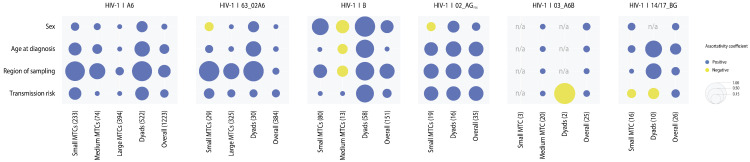
Assortativity within molecular transmission clusters by demographic and risk factors. Assortativity coefficients were calculated for each type of cluster as well as for all clusters combined (“overall”) within each of the major HIV-1 subtypes, based on four characteristics: sex (male, female), age at diagnosis (<35, 35–49, >50 years), region of sampling (subject-level), and transmission risk group (HET, IDUs, MSM, MTCT, NSC, and unknown). Coefficient values are displayed as bubbles; larger values indicate stronger assortativity. n/a, not applicable.

**Figure 5 pathogens-14-00738-f005:**
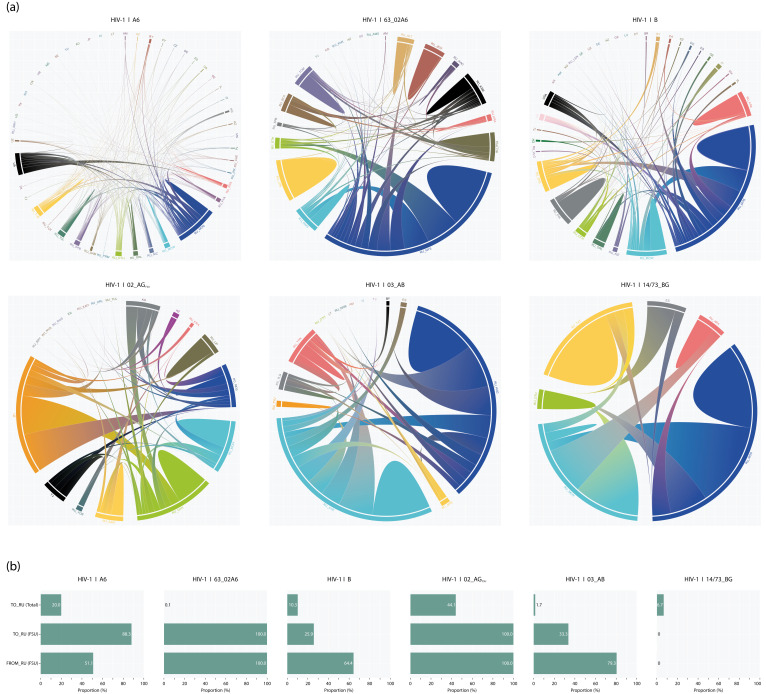
International and internal viral migration patterns of HIV-1 subtypes in Russia. (**a**) Sankey diagram illustrating inferred directional migration events for six major HIV-1 subtypes: A6, 63_02A6, B, 02_AG_FSU_, 03_A6B, and 14/73_BG. Arrows indicate source-to-recipient links; colors denote origin. Countries are labeled using ISO 3166-1 alpha-2 codes [46]; Russian regions use custom 3-letter codes. Only locations contributing >2.0% (≥3 sequences) of subtype-specific datasets are shown; all others are grouped as “other”. (**b**) Bar chart summarizing proportions of viral introductions to Russia and transmissions from Russia involving other Former Soviet Union (FSU) countries. Russian subject codes: ALT, Altai krai; BRY, Bryansk oblast; CHE, Chelyabinsk oblast; IRK, Irkutsk oblast; JEW, Jewish autonomous oblast; KAG, Kaliningrad oblast; KAO, Karachay-Cherkess republic; KEM, Kemerovo oblast; KHK, republic of Khakassia; KRA, Krasnodar krai; KYA, Krasnoyarsk krai; LEN, Leningrad oblast; LIP, Lipetsk oblast; MOS, Moscow oblast; MOW, Moscow (city); NGR, Novgorod oblast; NIZ, Nizhny Novgorod oblast; NVS, Novosibirsk oblast; OMS, Omsk oblast; ORL, Oryol oblast; OTH, Other; PRI, Primorsky krai; ROS, Rostov oblast; SAM, Samara oblast; SPB, Saint Petersburg (city); STA, Stavropol krai; SVE, Sverdlovsk oblast; TAT, republic of Tatarstan; TOM, Tomsk oblast; TUL, Tula oblast; TVE, Tver oblast; TYU, Tyumen oblast; VLG, Vologda oblast; VOR, Voronezh oblast; YAN, Yamalo-Nenets Autonomous okrug.

**Table 1 pathogens-14-00738-t001:** General characteristics of study participants stratified on HIV-1 subtypes.

	HIV-1 Subtype							
	Total (*n* = 9500, 100.0%)	A6 (*n* = 7659; 80.6% (95% CI, 79.8–81.4)	63_02A6 (*n* = 747; 7.9%; (95% CI, 7.3–8.4)	B (*n* = 531; 5.6% (95% CI, 5.1–6.1)	02_AG_FSU_ (*n* = 111; 1.2% (95% CI, 0.1–1.4)	03_A6B (*n* = 68; 0.7% (95% CI, 0.6–0.9)	14/73_BG (*n* = 59; 0.6% (95% CI, 0.5–0.8)	Others ^1^ (*n* = 325; 3.4% (95% CI, 3.1–3.8)
Age at diagnosis (in years)								
<20	1034 (10.9)	887 (11.6)	55 (7.4)	33 (6.2)	6 (5.4)	10 (14.7)	2 (3.4)	41 (12.6)
20–29	3406 (35.9)	2817 (36.8)	165 (22.1)	215 (40.5)	40 (36.0)	29 (42.6)	29 (49.2)	111 (34.1)
30–39	3084 (32.5)	2422 (31.6)	299 (40.0)	185 (34.8)	38 (34.2)	22 (32.4)	14 (23.7)	104 (32.0)
40–49	1336 (14.1)	1025 (13.4)	158 (21.2)	68 (12.8)	21 (18.9)	6 (8.8)	8 (13.6)	50 (15.4)
≥50	640 (6.7)	508 (6.6)	70 (9.4)	30 (5.6)	6 (5.4)	1 (1.5)	6 (10.2)	19 (5.8)
Median (IQR)	30.0 (24.0–38.0)	30.0 (23.0–37.0)	35.0 (28.0–41.0)	30.0 (25.0–38.0)	32.0 (25.0–39.0)	26.0 (22.0–34.0)	28.0 (24.0–38.0)	30.0 (24.0–37.0)
Sex								
Female	3864 (40.7)	3310 (43.2)	294 (39.4)	63 (11.9)	41 (36.9)	42 (61.8)	6 (10.2)	108 (33.2)
Male	5636 (59.3)	4349 (56.8)	453 (60.6)	468 (88.1)	70 (63.1)	26 (38.2)	53 (89.8)	217 (66.8)
Transmission risk								
HET	4140 (43.6)	3482 (45.5)	281 (37.6)	169 (31.8)	49 (44.1)	27 (39.7)	21 (35.6)	111 (34.1)
IDUs	2541 (26.7)	2195 (28.7)	213 (28.5)	38 (7.2)	19 (17.1)	12 (17.6)	2 (3.4)	62 (19.1)
MSM	535 (5.6)	226 (3.0)	5 (0.7)	202 (38.0)	19 (17.1)	1 (1.5)	22 (37.3)	60 (18.5)
MTCT	315 (3.3)	278 (3.6)	18 (2.4)	4 (0.8)	3 (2.7)	3 (4.4)	0 (0)	9 (2.8)
NSC	62 (0.7)	40 (0.5)	6 (0.8)	0 (0)	0 (0)	0 (0)	1 (1.7)	15 (4.6)
UNK	1275 (13.4)	985 (12.9)	140 (18.7)	60 (11.3)	14 (12.6)	22 (32.4)	6 (10.2)	48 (14.8)
ST	632 (6.7)	453 (5.9)	84 (11.2)	58 (10.9)	7 (6.3)	3 (4.4)	7 (11.9)	20 (6.2)
Diagnosis date								
before 2000	536 (5.6)	452 (5.9)	0 (0)	51 (9.6)	0 (0)	3 (4.4)	0 (0)	30 (9.2)
2001–2005	1188 (12.5)	1105 (14.4)	10 (1.3)	37 (7.0)	4 (3.6)	9 (13.2)	1 (1.7)	22 (6.8)
2006–2010	1902 (20.0)	1644 (21.5)	64 (8.6)	95 (17.9)	21 (19.0)	31 (45.6)	4 (6.8)	43 (13.2)
2011–2015	2214 (23.3)	1767 (23.0)	186 (24.9)	144 (27.1)	21 (18.9)	14 (20.6)	18 (30.5)	64 (19.7)
2016–2020	2270 (24.0)	1805 (23.6)	180 (24.1)	142 (26.7)	33 (29.7)	10 (14.7)	24 (40.7)	76 (23.4)
after 2020	1390 (14.6)	886 (11.6)	307 (41.1)	62 (11.7)	32 (28.8)	1 (1.5)	12 (20.3)	90 (27.7)

^1^ 01_AE (*n* = 10), 02_AG_African_ (*n* = 4), 06_cpx (*n* = 4), 11_cpx (*n* = 1), 18_cpx (*n* = 1), 19_cpx (*n* = 5), 20_BG (*n* = 3), 24_BG (*n* = 2), 141_BF1 (*n* = 1), A1 (*n* = 4), A7 (*n* = 3), C (*n* = 13), D (*n* = 2), F1 (*n* = 5), G (*n* = 18), URF_A6B (*n* = 92), non-A6B URFs (*n* = 70), 02_AG_African_-like (*n* = 3), 02_AG_FSU_-like (*n* = 12), 03_A6B-like (3), 63_02A6-like (*n* = 34), A6-like (*n* = 23), B-like (*n* = 3), C-like (*n* = 2) and G-like (*n* = 7). See Appendix A for a more complete description of these subtypes data. The sampling date is presented in Appendix A. IQR, interquartile range; CI, confidence interval; HET, heterosexual contacts; IDUs, injecting drug users; MSM, men who have sex with men; MTCT, mother-to-child transmission; NSC, nosocomial transmission; UNK, unknown; ST, sexual transmission (unspecified).

**Table 2 pathogens-14-00738-t002:** Multivariable analysis of selected demographic characteristics of study participants infected with subtype A6 and other HIV-1 subtypes.

Characteristics	Estimate	St. Error	t Value	*p* Value	Odds Ratio (95% CI)
Sex (female vs. male)	0.414	0.058	7.103	<0.001	1.51 (1.35–1.70)
Age at diagnosis (in years)	0.014	0.002	6.693	<0.001	3.60 (2.47–5.24)
Region of sampling (Central FD vs. Other) ^1^	0.221	0.054	4.076	<0.001	1.25 (1.12–1.39)
Transmission category					
HET vs. IDUs	−0.134	0.079	−1.700	0.089	0.87 (0.74–1.02)
HET vs. MSM	0.996	0.057	17.483	<0.001	7.33 (5.86–9.17)
HET vs. Other ^2^	0.119	0.024	5.032	<0.001	1.34 (1.24–1.64)

^1^ Federal District (FD) of Russia; Other—Far Eastern FD, North Caucasian FD, Northwestern FD, Siberian FD, Southern FD, Ural FD and Volga FD, in the aggregate; ^2^ Mother-to-child transmission, nosocomial transmission, sexual transmission (without specification) and participants with unknown risk of transmission, in the aggregate. CI, Confidence interval; FD, Federal District; HET, heterosexual contacts; IDUs, injecting drug users; MSM, men who have sex with men.

**Table 3 pathogens-14-00738-t003:** Factors associated with clustering among HIV-positive individuals within transmission networks.

Attribute	Category	Total, *n*	Non-Clustering, *n* (%)	Clustering, *n* (%)	Adjusted Odds Ratio (95% CI)	*p*-Value
Age at diagnosis (in years)						
	≥50	621	504 (81.2)	117 (18.8)	Ref	
	30–49	4266	3400 (79.7)	866 (20.3)	1.17 (0.93–1.46)	0.180
	<30	4288	3427 (79.9)	861 (20.1)	1.31 (1.03–1.66)	0.028
Sex						
	Female	3756	3023 (80.5)	733 (19.5)	Ref	
	Male	5419	4308 (79.5)	1111 (20.5)	1.09 (0.97–1.22)	0.143
Transmission risk						
	HET	4029	3247 (80.6)	782 (19.4)	Ref	
	IDUs	2479	1954 (78.8)	525 (21.2)	1.28 (1.01–1.62)	0.043
	MSM	475	338 (71.2)	137 (28.8)	1.11 (0.86–1.44)	0.410
	MTCT	306	254 (83.0)	52 (17.0)	1.35 (0.96–1.90)	0.083
	NSC	47	21 (44.7)	26 (55.3)	5.26 (2.83–9.76)	<0.001
	UNK	1228	1003 (81.7)	225 (18.3)	0.83 (0.70–1.00)	0.054
	ST	611	514 (84.1)	97 (15.9)	0.62 (0.48–0.82)	0.001
ART status						
	Treated	5084	4414 (86.8)	670 (13.2)	Ref	
	Naive	4091	2917 (71.3)	1174 (28.7)	2.70 (2.40–3.04)	<0.001
HIV-1 subtype						
	A6	7659	6436 (84.0)	1223 (16.0)	Ref	
	B	531	380 (71.6)	151 (28.4)	2.16 (1.76–2.66)	<0.001
	63_02A6	747	363 (48.6)	384 (51.4)	5.70 (4.82–6.73)	<0.001
	02_AG_FSU_	111	76 (68.5)	35 (31.5)	2.37 (1.56–3.60)	<0.001
	03_A6B	68	43 (63.2)	25 (36.8)	2.81 (1.68–4.70)	<0.001
	14/73_BG	59	33 (55.9)	26 (44.1)	3.54 (2.06–6.06)	<0.001
Region of sampling ^1^						
	CN	3840	3057 (79.6)	783 (20.4)	Ref	
	FE	411	331 (80.5)	80 (19.5)	0.84 (0.64–1.09)	0.193
	NC	267	182 (68.2)	85 (31.8)	1.81 (1.35–2.43)	<0.001
	NW	834	787 (94.4)	47 (5.6)	0.28 (0.20–0.39)	<0.001
	SB	1111	794 (71.5)	317 (28.5)	1.31 (1.10–1.56)	0.002
	ST (FD)	870	576 (66.2)	294 (33.8)	1.55 (1.31–1.84)	<0.001
	UR	764	667 (87.3)	97 (12.7)	0.49 (0.39–0.63)	<0.001
	VL	1078	937 (86.9)	141 (13.1)	0.62 (0.50–0.75)	<0.001
Diagnosis date						
	1988–2004	1476	1192 (80.8)	284 (19.2)	Ref	
	2005–2015	4205	3478 (82.7)	727 (17.3)	0.60 (0.51–0.72)	<0.001
	2016–2023	3494	2661 (76.2)	833 (23.8)	0.76 (0.61–0.93)	0.009

^1^ Federal District (FD) of Russia; CI, confidence interval; HET, heterosexual contacts; IDUs, injecting drug users; MSM, men who have sex with men; MTCT, mother-to-child transmission; NSC, nosocomial transmission; UNK, unknown; ST, sexual transmission; CN, Central FD; FE, Far East FD; NC, North Caucasian FD; NW, Northwestern FD; SB, Siberian FD; ST (FD), Southern FD; UR, Ural FD; VL, Volga FD.

**Table 4 pathogens-14-00738-t004:** Characteristics of HIV-1 phylogenetic clusters (PCs) identified in Russia.

Subtype	Phylogenetic Cluster (Subcluster, #)	Size, *n* ^1^	Sampling Year	Likely Phylogenetic Origin	Geographic Location (Proportion, %) ^2^	TMRCA (95% CI)
A6	Overall	12,396	1997–2024	Ukraine	Mixed	1993.2 (1992.0–1994.9)
	Cluster 1	540	2001–2023	Russia, Moscow oblast	Mixed [Russian (76.1)]	1996.8 (1995.9–1998.3)
	Cluster 2	203	2007–2022	Belarus	Belarus (99.5) [Russian (0.5)]	1999.1 (1997.8–2000.6)
	Cluster 3	140	2011–2023	Poland	Poland (86.4%) [Russian (3.6)]	2002.8 (2001.6–2004.1)
	Cluster 4	101	2010–2021	Ukraine	Poland (95.0) [Russian (1.0)]	2005.6 (2003.8–2007.4)
	Cluster 5	93	2005–2023	Ukraine	Krasnoyarsk krai (78.5) [Russian (94.6)]	1999.1 (1997.6–2000.8)
	Cluster 6	84	2005–2014	Ukraine	Latvia (88.1) [Russian (3.6)]	2000.1 (1998.8–2001.6)
	Cluster 7	61	2018–2023	Ukraine	Orel oblast (98.4) [Russian (100.0)]	2004.4 (2002.7–2006.1)
	Cluster 8	55	2002–2022	Ukraine	Mixed [Russian (81.8)]	2000.0 (1998.4–2001.5)
B	Overall	3034	1978–2024	United States	Mixed	1972.5 (1967.5–1976.1)
	Cluster 1	192	1995–2023	United States	Mixed [Russian (72.4)]	1978.1 (1977.3–1978.8)
	#1.1	33	1995–2023	Russia, Moscow (city)	Mixed [Russian (85.3)]	1982.7 (1981.7–1984.1)
	#1.2	16	1995–2023	Russia, St. Petersburg (city)	Mixed [Russian (93.7)]	1988.3 (1986.8–1990.2)
	#1.3	15	2006–2017	Russia, Moscow oblast	Krasnodar krai (73.3) [Russian (100.0)]	1987.4 (1985.6–1989.5)
	Cluster 2	273	1987–2023	Poland	Mixed [Russian (20.5)]	1982.9 (1980.8–1985.2)
	#2.1	106	1996–2023	Poland	Mixed [Russian (51.9)]	1988.7 (1987.4–1990.3)
	Cluster 3	87	1995–2023	Russia, Moscow oblast	Mixed [Russian (87.3)]	1981.8 (1982.0–1982.8)
	#3.1	29	1995–2022	Russia, Moscow (city)	Mixed [Russian (93.1)]	1987.2 (1985.8–1988.8)
	#3.2	32	1999–2023	Russia, Moscow oblast	Mixed [Russian (96.9)]	1984.3 (1983.3–1985.4)
	Cluster 4	80	1995–2023	Russia, Moscow oblast	Mixed [Russian (66.2)]	1980.9 (1979.6–1982.7)
	Cluster 5	70	1995–2023	Russia, Moscow oblast	Mixed [Russian (68.6)]	1983.0 (1982.0–1984.5)
	#5.1	24	2004–2020	Russia, Moscow oblast	Mixed [Russian (54.2)]	1987.0 (1985.6–1988.8)
	#5.2	14	2008–2023	Russia, Moscow oblast	Mixed [Russian (85.7)]	1988.7 (1986.5–1991.4)
	Cluster 6	46	2003–2021	Italy	Mixed [Russian (43.5)]	1986.8 (1985.2–1988.8)
	#6.1	21	2009–2020	Italy/Russia, Moscow oblast	Mixed [Russian (90.5)]	1992.3 (1990.4–1994.3)
	Cluster 7	41	1995–2023	Russia, Moscow oblast	Mixed [Russian (85.4)]	1985.7 (1981.0–1986.7)
	Cluster 8	17	2007–2020	United States/Russia, St. Petersburg (city)	Mixed [Russian (76.5)]	1990.5 (1988.6–1993.1)
	Cluster 9	14	2002–2022	United States/Russia, St. Petersburg (city)	Mixed [Russian (50.0)]	1985.8 (1984.7–1987.7)
	Cluster 10	12	1999–2021	United Kingdom	Mixed [Russian (63.6)]	1987.3 (1985.5–1989.0)
63_02A6	Overall	1372	2008–2024	Russia, Novosibirsk oblast	Mixed	2004.3 (2002.8–2005.7)
	Cluster 1	129	2016–2023	Russia, Tomsk oblast	Tomsk oblast (44.2) [Russian (99.2)]	2010.6 (2009.9–2011.6)
	#1.1	62	2016–2023	Russia, Stavropol krai/Krasnodar krai	Karachay-Cherkess rep. (46.8) [Russian (100.0)]	2012.2 (2011.7–2012.8)
	Cluster 2	108	2016–2023	Russia, Novosibirsk oblast	Orel region (98.1) [Russian (100.0)]	2012.1 (2011.2–2013.1)
	Cluster 3	92	2011–2023	Russia, Kemerovo oblast	Kemerovo oblast (37.0) [Russian (100.0)]	2009.3 (2008.7–2009.9)
	Cluster 4	54	2015–2023	Russia, Novosibirsk oblast	Krasnoyarsk krai (48.1) [Russian (100.0)]	2010.9 (2010.3–2012.1)
	Cluster 5	37	2015–2024	Russia, Novosibirsk oblast	Jewish Aut. oblast (97.3) [Russian (100.0)]	2010.7 (2010.1–2011.6)
	Cluster 6	31	2015–2023	Russia, Novosibirsk oblast	Mixed [Russian (100.0)]	2008.8 (2008.4–2009.3)
02_AG_FSU_	Overall	309	2002–2023	Uzbekistan	Mixed	1999.6 (1997.2–2001.8)
	Cluster 1	25	2014–2023	Russia, Moscow (city)	Mixed [Russian (96.0)]	2005.1 (2004.0–2007.3)
	Cluster 2	24	2009–2021	Uzbekistan	Kyrgyzstan (75.0) [Russian (25.0)]	2006.7 (2005.7–2008.0)
	Cluster 3	9	2015–2023	Uzbekistan	Lipetsk oblast (77.8) [Russian (100.0)]	2006.0 (2004.6–2007.7)
	Cluster 4	7	2018–2022	Kazakhstan/Kyrgyzstan	Moscow (city) (57.1) [Russian (85.7)]	2010.4 (2008.3–2012.9)
	Cluster 5	6	2017–2023	Kyrgyzstan	Kyrgyzstan (83.3) [Russian (16.7)]	2012.6 (2011.0–2014.3)
03_A6B	Overall	300	1997–2024	Russia, Kaliningrad oblast	Mixed	1990.8 (1987.6–1994.0)
	Cluster 1	290	1997–2024	Russia, Kaliningrad oblast	Mixed [Russian (88.6)]	1993.4 (1991.1–1995.9)
	Cluster 2	10	1998–2017	Russia, Kaliningrad oblast	Mixed [Russian (100.0)]	1996.3 (1993.9–1997.0)
14/73_BG	Overall	143	1998–2023	Spain	Mixed	1991.6 (1987.3–1993.3)
	Cluster 1	56	2009–2023	Spain	Mixed [Russian (100.0)]	1999.9 (1998.1–2002.3)
	#1.1	10	2010–2022	Russia, Moscow (city)	Moscow (city) (40.0%) [Russian (100.0)]	2008.0 (2006.8–2009.3)
	#1.2	14	2009–2022	Russia, Moscow (city)/Moscow oblast	Tatarstan (71.4%) [Russian (100.0)]	2003.3 (2001.4–2005.2)
	Cluster 2	3	2008–2023	Spain	Bashkortostan (66.7%) [Russian (100.0)]	2005.8 (2004.0–2008.0)

^1^ Only large (relatively) PCs consisting of at least 2.0% of all sequences within a specific HIV-1 subtype/CRFs are described in the table; ^2^ Only locations (country and/or subject of Russia) accounting for >35.0% of all sequences within a specific PCs are described in the table; PCs made with less than 50% participation by any one location (country and/or subject of Russia) were considered mixed. The total share of Russian sequences is included in square brackets.

## Data Availability

The data presented in this study are available on request from the corresponding author. The data are not publicly available due to privacy policy of HIV Russian database.

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
