# Peer review of "The Molecular Epidemiology of HIV-1 in Russia, 1987–2023: Subtypes, Transmission Networks and Phylogenetic Story"

_pathogens, 2025, doi:10.3390/pathogens14080738_

Round 1

Reviewer 1 Report

Comments and Suggestions for Authors

I have examined the manuscript entitled " The molecular epidemiology of HIV-1 in Russia, 1987-2023: 2 Subtypes, Transmission Networks and Phylogenetic Story" authored by Aleksey Lebedev and colleagues.

The manuscript presents a comprehensive molecular epidemiological analysis of HIV-1 in Russia, representing a groundbreaking effort to address significant gaps in national HIV surveillance. In an era when understanding viral transmission networks is crucial for epidemic control, this project provides a timely response by analyzing the largest national HIV-1 dataset from Russia spanning three decades (1987-2023). The authors have created a robust framework that not only compiles genetic sequences across all federal districts but also integrates phylogenetic analysis with epidemiological data to offer unprecedented insights into transmission dynamics.

The Russian HIV-1 epidemic presents unique challenges that make this study particularly significant. As the largest country in the world, Russia's vast territory spans multiple time zones and diverse populations, creating complex transmission dynamics that differ markedly from smaller, more homogeneous settings. With over 1.1 million officially registered HIV cases as of 2023, Russia faces one of the most substantial HIV epidemics globally. The epidemic is further characterized by the dominance of subtype A6, a variant that entered Russia via Ukraine in the mid-1990s and rapidly spread among injection drug users before expanding into the general population. The specific epidemiological patterns and geographic distribution of this subtype across Russia's federal districts create a unique molecular landscape that necessitates specialized surveillance approaches, rendering this comprehensive analysis particularly valuable for understanding regional epidemic dynamics.

One of the most notable strengths of this study is its direct relevance to public health policy. By focusing on HIV-1, a global health priority requiring targeted interventions, the authors address an urgent need for evidence-based epidemic control strategies. The study's capability to identify active transmission clusters, track the evolution from injection drug use to sexual transmission, and monitor the geographic spread of different subtypes showcases its practical utility for policymakers and public health officials. The inclusion of sophisticated visualization tools and network analysis further elevates its value, converting complex molecular data into actionable insights for intervention planning.

From a methodological standpoint, the paper excels in rigor and reproducibility. The authors thoroughly outline their phylogenetic reconstruction methods, transmission network analysis, and statistical approaches, incorporating established tools, and comprehensive quality control measures. This methodological transparency ensures that other researchers can replicate or build upon their findings. Furthermore, their integration of multiple analytical approaches, combining phylodynamic analysis, network reconstruction, and epidemiological modeling, sets an exemplary standard for molecular epidemiological research.

The authors' approach is further highlighted by their discussion of how this molecular surveillance framework could enhance existing HIV monitoring systems, providing valuable context regarding its potential integration within broader public health infrastructure and its complementary role in epidemic control efforts.

In summary, this manuscript presents a meticulously crafted and impactful analysis that significantly advances the field of HIV molecular epidemiology. Its focus on comprehensive national surveillance, real-time transmission monitoring, and integration of molecular and epidemiological data positions it as a major contribution to the field. The work represents a considerable advancement in understanding HIV-1 dynamics in Russia and could serve as a model for similar surveillance efforts in other regions.

Author Response

Dear Reviewer 1,

The authors express their deep gratitude for your positive evaluation of our work and the kind words about the manuscript. We will strive to maintain the same level of research and publication quality in our future work.

Reviewer 2 Report

Comments and Suggestions for Authors

The authors performed a comprehensive and large-scale investigation of the HIV-1 epidemic in Russia ranging over three decades (time range from 1987 till 2023), hence delivering one of the most complete national-level analyses to date. The size of the dataset analyzed (9500 cases), although only comprising sequences from 1% of the HIV-infected Russian population, was big enough to ensure robust integration of phylogenetic, epidemiological, demographic, and behavioral data. Moreover, network analysis was performed to reveal the dynamics of molecular transmission clusters (MTCs).

The findings of this research highlight that subtype A6 remains the dominant subtype nationwide, accounting for over 80% of the infections. Nevertheless, the prevalence of non-A6 subtypes (such as 63_02A6, subtype B, 02_AGFSU, 03_A6B, and 14/73_BG) is increasing and signals a clear shift toward greater viral diversity within the population. This diversification is both linked to local as well as cross-border transmission dynamics, with non-A6 variants having a higher chance to form active clusters, especially among key risk groups such as men who have sex with men, injecting drug users, and individuals not receiving antiretroviral therapy.

The authors discussion was interesting and complete, including as well the drawbacks and limitations of their research. Altogether, the results presented in this manuscript described the evolution of the HIV-1 epidemy in Russia and underscore the critical role of continued molecular surveillance in informing public health strategies.

Revisions to be considered:

°In the cluster analysis clusters are defined in size ranging from 2 till 394 participants (line 315) and general info concerning the clustered patients do include all of these.  On line 318 a distinction is made for participants in dyads compared to participants in small, medium and large clusters. It would be more accurate and clear to the reader to describe the dyads as a separate group next to the singletons and the clustered patients. Especially as characteristics attributed to dyads are often different to those seen in clusters and singletons.

°The legend of figure 3 is not correct. What is described as panel b) does not correspond to what is seen in the figures.

°Also in figure 3: The legends attached to the figures describing the subtypes contain errors: The upper left figure does only describe subtypes A6 and 63_02A6 and not the remaining subtypes. The figure just thereunder is the opposite. Nevertheless all subtypes are mentioned in the legends of these separate figure. Please adapt accordingly.

°Tabel 1: mentions the sampling year with only 4 samples from before 2005. The text also mentions that samples were collected from 1995 onwards, but it should be mentioned that only 1 sample was from before 2000. Also the paper states that it describes the epidemiology from 1987 till 2023 in Russia. But 99% of the samples were only collected from 2006 onwards. Would this discrepancy have any effect on the results described? Should be elaborated in de discussion.

°The text in the figures are very blurry and hence unreadable even when zooming in. This is annoying in many figures but especially in figure 5 were it is unreadable and not to be determined based on what is described in the legend.

°This part of the legend in figure 4 is difficult to interpret: “Assortativity coefficients were calculated for each major HIV-1 subtype and for all clusters combined ("overall"),”. What you mean is that assortativity coefficients were calculated for each type of cluster as well as for all clusters combined (“overall”) and this for each of the major HIV-1 subtypes. This phrase should hence be adapted to make it correct.

Author Response

Dear Reviewer 2,

The authors express their sincere gratitude for your careful review of our article and for the valuable and fair comments provided. We have made every effort to address all suggestions and revisions, and we hope that the manuscript is now clearer and better formulated.

Please find my detailed replies below.

Comments 1: In the cluster analysis clusters are defined in size ranging from 2 till 394 participants (line 315) and general info concerning the clustered patients do include all of these.  On line 318 a distinction is made for participants in dyads compared to participants in small, medium and large clusters. It would be more accurate and clear to the reader to describe the dyads as a separate group next to the singletons and the clustered patients. Especially as characteristics attributed to dyads are often different to those seen in clusters and singletons.

Response 1:  Agree. When highlighting a separate group of dyads as a form of MTCs, we were guided by the understanding that the characteristics inherent in dyads (pairs of individuals representing a potential event of direct transmission) may differ from those observed in larger clusters, as you rightly pointed out. That is why such comparisons are made (Fig. 3b, Fig. 4, and the main text). As for the characteristics associated with singletons, we present them here simply as additional information (Table S2).

Comments 2: The legend of figure 3 is not correct. What is described as panel b) does not correspond to what is seen in the figures.

Response 2: Thank you for pointing this out. We agree with this comment. Therefore, we have a correction (addition).

Namely: «Figure 3. Molecular transmission clusters (MTCs) among individuals with HIV-1 in Russia. (a) Visualization of selected MTCs with node attributes showing subtype, age at diagnosis, transmission risk, region (FD), and sex. Each node represents a person; edges indicate pairwise genetic distances ≤0.75% for A6 and 63_02A6 (top panel), and ≤1.5% for subtype B, 02_AGFSU, CRF03_A6B, and 14/73_BG (bottom panel). (b) Clustering characteristics of four types of clusters involving a major HIV-1 subtypes in the network. HET, heterosexual contact; IDUs, injecting drug users; MSM, men who have sex with men; MTCT, mother-to-child transmission; NSC, nosocomial transmission; UNK, unknown; SXL, sexual transmission (unspecified); CN, Central FD; FE, Far Eastern FD; NC, North Caucasian FD; NW, Northwestern FD; SB, Siberian FD; ST, Southern FD; UR, Ural FD; VL, Volga FD; LMTCs, large MTCs; MMTCs, medium MTCs; SMTCs, small MTCs.»

Page 12, legend of Figure 3, lines 368-372

Comments 3: Also in figure 3: The legends attached to the figures describing the subtypes contain errors: The upper left figure does only describe subtypes A6 and 63_02A6 and not the remaining subtypes. The figure just thereunder is the opposite. Nevertheless, all subtypes are mentioned in the legends of these separate figure. Please adapt accordingly.

Response 2: Agree. We have, accordingly, changed in Figure 3. Now only the subtypes A6 and 63_02A6 are mentioned in the captions for the upper left figure, while other subtypes are mentioned in the captions for the lower figure.

Page 12, Figure 3

Comments 4: Tabel 1: mentions the sampling year with only 4 samples from before 2005. The text also mentions that samples were collected from 1995 onwards, but it should be mentioned that only 1 sample was from before 2000. Also the paper states that it describes the epidemiology from 1987 till 2023 in Russia. But 99% of the samples were only collected from 2006 onwards. Would this discrepancy have any effect on the results described? Should be elaborated in de discussion.

Response 4: This discrepancy would not have a significant impact on the results we described, because most of them (with the exception of active cluster detection and phylodynamic analysis) used information on the date of diagnosis rather than the date of sampling. Based on the assumption that a person in a productive infection state is a carrier of an established HIV-1 subtype (even if initially carrying mixed HIV-1 infections) and the subtype will not change over time, this approach seems correct to us. Regarding the phylodynamic analysis, sufficient temporal structure to reliably estimate the evolutionary rate and divergence time of clusters has been confirmed by examining temporal signal and “clocklikeness ”. However, your question prompted us to change this column.  Accordingly, we changed the “Samping Year” to “Diagnosis Date” information to emphasize this point; we moved the sample year information to a supplementary table.

Page 6-7, Table 1.

We also modified the “Overview of data” paragraph to add data on diagnosis date. Namely: «Based on these criteria, 9,500 patients with HIV-1 diagnosed between 1988 and 2023 from all 8 Federal 135 Districts (FDs) and 78 of Russia’s 89 federal subjects (87.6%) were included; virologic data (genomes) were obtained from 1995 to 2023»

Page 3, paragraph 2.2, lines 135-137

Comments 5: The text in the figures are very blurry and hence unreadable even when zooming in. This is annoying in many figures but especially in figure 5 were it is unreadable and not to be determined based on what is described in the legend.

Response 5: Agree. It seems that the quality of drawings decreased during the conversion process.  Initially all drawings are high resolution (1200 dpi) and do not lose quality when scaled (up to 400%). We will draw the editor's attention to this.

Comments 6: This part of the legend in figure 4 is difficult to interpret: “Assortativity coefficients were calculated for each major HIV-1 subtype and for all clusters combined ("overall"),”. What you mean is that assortativity coefficients were calculated for each type of cluster as well as for all clusters combined (“overall”) and this for each of the major HIV-1 subtypes. This phrase should hence be adapted to make it correct.

Response 6: Agree. We have, accordingly, changed of the legend in figure 4. Namely: «Assortativity coefficients were calculated for each type of cluster as well as for all clusters combined (“overall”) within each of the major HIV-1 subtypes.»

Page 13, legend of Figure 4, lines 393-394